# Clinicopathological Features of Gastric Cancer with Autoimmune Gastritis

**DOI:** 10.3390/biomedicines10040884

**Published:** 2022-04-12

**Authors:** Junya Arai, Ryota Niikura, Yoku Hayakawa, Nobumi Suzuki, Yoshihiro Hirata, Tetsuo Ushiku, Mitsuhiro Fujishiro

**Affiliations:** 1Department of Gastroenterology, Graduate School of Medicine, The University of Tokyo, Tokyo 113-8655, Japan; jarai-tky@umin.ac.jp (J.A.); nsuzuki-ham@umin.ac.jp (N.S.); yohirata@ims.u-tokyo.ac.jp (Y.H.); fujishiromi-int@h.u-tokyo.ac.jp (M.F.); 2Department of Gastroenterological Endoscopy, Tokyo Medical University, Tokyo 160-0023, Japan; 3Department of Pathology, Graduate School of Medicine, The University of Tokyo, Tokyo 113-8655, Japan; ushikut-pat@h.u-tokyo.ac.jp

**Keywords:** autoimmune gastritis, gastric cancer, mucin

## Abstract

Most gastric cancers develop in patients with chronic gastritis. Chronic gastritis can be classified into two major subtypes: *Helicobacter pylori* (*H. pylori*)-induced gastritis and autoimmune gastritis (AIG). Whereas *H. pylori*-related gastric cancers are more common and have been extensively investigated, the clinicopathological features of gastric cancer with autoimmune gastritis are unclear. Patients diagnosed with gastric cancer and hospitalized in the University Tokyo Hospital from 1998 to 2017 were enrolled. Diagnosis of autoimmune gastritis was based on positivity for serum anti-parietal cell antibody (APCA). We evaluated mucin expression and immune cell infiltration by immunohistochemical staining for MUC5AC, MUC6, PD-L1, CD3, CD11, Foxp3, and PD1. We also examined the presence of bacterial taxa that are reportedly enriched in AIG. Survival analyses of recurrence and 5-year mortality were also performed. In total, 261 patients (76 APCA-positive and 185 APCA-negative) were analyzed. Immunohistochemical staining in the matched cohort showed that AIG-related gastric cancer had higher MUC5AC expression (*p* = 0.0007) and MUC6 expression (*p* = 0.0007). Greater infiltration of CD3-positive (*p* = 0.001), Foxp3-positive (*p* < 0.001), and PD1-positive cells (*p* = 0.001); lesser infiltration of CD11b-positive (*p* = 0.005) cells; and a higher prevalence of Bacillus cereus (*p* = 0.006) were found in AIG-related gastric cancer patients. The cumulative incidences of gastric cancer recurrence were 2.99% at 2 years, 15.68% at 6 years, and 18.81% at 10 years in APCA-positive patients; they were 12.79% at 2 years, 21.35% at 6 years, and 31.85% at 10 years in APCA-negative patients. The cumulative incidences of mortality were 0% at 3 years and 0% at 5 years in APCA-positive patients; they were 1.52% at 3 years and 2.56% at 5 years in APCA-negative patients. We identified molecular differences between AIG and non-AIG gastric cancer. Differences in T-cell populations and the gastric microbiota may contribute to the pathogenesis of gastric cancers and potentially affect the response to immunotherapy.

## 1. Introduction

Chronic gastritis is a risk factor for gastric cancer. There are two major types of chronic gastritis: *Helicobacter pylori* (*H. pylori*)-related gastritis and autoimmune gastritis (AIG). *H. pylori*-related gastritis occurs after chronic *H. pylori* infection in the stomach. On the other hand, AIG-related gastritis is the result of an autoimmunological attack by T cells that recognize acid-producing parietal cells in the proximal stomach; it can be diagnosed based on the serum anti-parietal cell antibody (APCA) status [1]. Both types of gastritis cause gastric atrophy and intestinal metaplasia, in which cells of the normal gastric lineage disappear and intestinal cell types emerge. The presence and severity of gastric atrophy and intestinal metaplasia are closely associated with gastric cancer risk [2,3,4,5,6]. However, the differences in cancer risk and pathogenesis between these two subtypes are unclear.

In AIG, parietal cells are destroyed by migrated cytotoxic T cells and, thus, gastric atrophy is more prominent in the proximal stomach, while the gastric mucosa usually remains intact in the distal stomach where no parietal cells exist [7]. In contrast, *H. pylori*-induced gastric atrophy and metaplasia initially occur at the distal stomach and gradually expand towards the proximal stomach. The destruction of parietal cells in AIG can be modeled by the TxA23 transgenic mouse model that overexpresses the T-cell receptor in parietal cells; the infiltration of T lymphocytes causes chronic inflammation in the proximal stomach [8]. *H. pylori*-infected mice develop pangastritis, accompanied by the robust recruitment of neutrophils and myeloid cells in the early stages [9] and modest degrees of T- and B-lymphocyte infiltration in the later stages [10].

Chronic inflammation promotes cancer development in various organs such as the stomach, liver, esophagus, colon, pancreas, uterus, and lung. Although cytotoxic immune cells (e.g., natural killer and CD8+ T cells) are expected to recognize and eliminate cancer cells that express atypical host antigens [11], cancer cells possess a mechanism for escaping surveillance by cytotoxic immune cells. Cancer cells and several immune cells express or secrete immunosuppressive molecules that inhibit the anti-tumor function of cytotoxic lymphocytes. For instance, myeloid-derived suppressor cells and regulatory CD4+ T cells (Tregs), which are commonly recruited in the tumor microenvironment, are responsible for suppressing the priming, activation, and cytotoxicity of effector cells [12]. Their immunosuppressive functions are regulated by cell–cell contact-dependent mechanisms such as the expression of PD-L1 or PD1 [13]. Recent studies have indicated the potential carcinogenesis of immune evasion such as that gained when the PD1/PD-L1 suppresses effector killer T cells [14] and activates regulatory T cells. PD-L1 is reportedly expressed in approximately half of gastric cancers; its expression on cancer cells is induced by interferon γ from T cells [15]. The distinct immune reactions of AIG and *H. pylori* gastritis may influence tumor immunity, along with the subsequent development and prognosis of gastric cancer.

To address these issues, we performed a retrospective cohort study in which we analyzed differences between AIG and *H. pylori*-related gastritis in terms of the clinicopathological features of gastric cancer, including the tumor immune system, mucin expression, and the gastric microbiota.

## 2. Materials and Methods

### 2.1. Study Design and Setting

We retrospectively collected data regarding consecutive hospitalized patients that had been diagnosed with gastric cancer at Tokyo University Hospital from 1998 to 2017. This cohort study protocol was approved by the Institutional Review Board of the University of Tokyo Hospital (no. 2058-(2)).

### 2.2. APCA Positivity and Pure AIG Definition

APCA positivity was defined as an APCA serum level >10 times. We compared the clinicopathological features of the APCA-positive and -negative groups. We defined pure AIG as follows: APCA positivity, endoscopy-confirmed gastric atrophy at the fundic gland zone, a gastrin serum level >200 pg/mL, and *H. pylori* negativity.

### 2.3. Patients

Patients pathologically diagnosed with gastric cancer based on biopsy findings were eligible for inclusion in the study. For immunohistochemical staining, we used another cohort of pure AIG and *H. pylori*-positive/APCA-negative patients identified by propensity score matching according to patient age and sex, as well as the position, differentiation, and stage of gastric cancer (Figure 1).

### 2.4. Outcomes and Variables

The primary outcomes were gastric cancer recurrence and 5-year mortality. Gastric cancer recurrence included both metachronous and metastatic ones in the stomach and the other organs. The following clinicopathological features of gastric cancer were analyzed: tumor position, Epstein–Barr virus status, differentiation, early/advanced cancer, staging, and initial treatment.

We also evaluated the following clinical factors: age, sex, smoking, *H. pylori* status, medication use (aspirin, proton pump inhibitor, histamine-2 receptor blocker, statin, nonsteroidal anti-inflammatory drugs, angiotensin-converting enzyme blocker, angiotensin II receptor blocker, α-blocker, β-blocker, vitamin B12, vitamin C, and metformin), and comorbidities (acquired immunodeficiency syndrome, cerebrovascular disease, chronic heart failure, chronic kidney disease, dementia, diabetes mellitus, hemiplegia, ischemic heart disease, liver cirrhosis, peripheral vascular disease, chronic obstructive pulmonary disease, rheumatic disease, peptic ulcer disease, leukemia, and lymphoma). *H. pylori* status was defined as the latest result from serological testing, a urea breath test, or a stool antigen test; it was categorized as positive, eradicated, negative, or unknown.

### 2.5. Immunohistochemical Staining of Mucin, PD-L1, CD3, CD11b, Foxp3, PD1, and Bacteria

Three-millimeter-thick sections were deparaffinized, rehydrated in phosphate-buffered saline, placed in 10 mM citrate buffer (pH 6.0), and heated to 120 °C for 5 min to recover antigenicity. Sections were preincubated with blocking buffer (3% hydrogen peroxide) for 5 min at room temperature. The primary anti-PD-L1 monoclonal antibody (1:100; cat. no. 13684; Cell Signaling Technology, Danvers, MA, USA) was diluted in phosphate-buffered saline and incubated with the sections overnight at 4 °C. Sections were then washed in phosphate-buffered saline and incubated with Histofine Simple Stain Max-Po (Multi) (Nichirei Biosciences Inc., Tokyo, Japan). After development with 3,3,9-diaminobenzidine tetrahydrochloride (DAB Substrate Kit, Nichirei Biosciences), sections were counterstained with hematoxylin and viewed under a light microscope. PD-L1 expression was assessed by one researcher (JA) using the tumor proportion score (TPS), which was defined as positive PD-L1 immunostaining in tumor cells and classified into the following three groups: TPS > 50%, 50% > TPS > 1%, and TPS < 1%. For immunofluorescence, the paraffin-embedded sections were incubated with primary antibodies against MUC2 (cat. no. ab90007; Abcam, Cambridge, UK), MUC5AC (cat. no. ab3649; Abcam), MUC6 (cat. no. NBP2-44374; Novusbio, Littleton, CO, USA), CD3 (cat. no. NCL-L-CD3-565; Leika, Wetzlar, Germany), CD11b (cat. no. 14011282; Invitrogen, Waltham, MA, USA), Foxp3 (cat. no. 98377; Cell Signaling Technology), PD1 (cat. no. 86163; Cell Signaling Technology), Bacillus cereus (cat. no. ab20556; Abcam), and Streptococcus Group B (cat. no. ab53584; Abcam). They were then incubated with secondary Alexa555 or Alexa488 IgG antibody (Invitrogen). CD3, CD11b, Foxp3, and PD1 expression patterns were assessed using the numbers of CD3-, CD11b-, Foxp3-, and PD1-expressing cells per visual field. B. cereus and Streptococcus Group B abundances were estimated using three grades.

### 2.6. Statistical Analysis

We performed one-to-one propensity score matching analysis between pure AIG and *H. pylori*-positive/APCA-negative patients using estimated propensity scores. To estimate the propensity score, we fitted a logistic model for pure AIG patients as a function of age, sex, position, differentiation, and staging. We calculated the c-statistic to evaluate the goodness-of-fit. Each pure AIG patient was matched with an APCA-negative/*H. pylori*-positive patient. Furthermore, patients were matched to patients with the most similar specified range (≤0.2 of the pooled standard deviation of estimated logits) to reduce differences between the two groups.

Continuous data were compared by *t*-test. Comparisons of categorical data between groups were performed by the chi-squared test or Fisher’s exact test as appropriate. The primary endpoints, 5-year mortality, and recurrence of gastric cancer, were censored at the date of the final visit. The Kaplan–Meier method was used to calculate the cumulative probabilities of the primary outcomes. *p*-values < 0.05 were considered indicative of statistical significance. Statistical analysis was performed using SAS software (ver. 9.4; SAS Institute, Cary, NC, USA).

## 3. Results

### 3.1. Patient Characteristics

In total, 261 gastric cancer patients (76 APCA-positive and 185 APCA-negative) were analyzed. Their characteristics are shown in Table 1 and Appendix A. The APCA-positive group showed higher proportions of elderly and chronic heart failure patients than did the APCA-negative group.

### 3.2. Primary Outcomes and Clinicopathological Features of Gastric Cancer

Primary outcomes and clinicopathological features associated with APCA are shown in Table 2 and Figure 2. Gastric cancers with APCA were associated with the early stages (*p* = 0.027). The mean follow-up period was 5.97 years. The cumulative incidences of gastric cancer recurrence were 2.99% at 2 years, 15.68% at 6 years, and 18.81% at 10 years in APCA-positive patients; they were 12.79% at 2 years, 21.35% at 6 years, and 31.85% at 10 years in APCA-negative patients (Figure 1). APCA positivity was not significantly associated with the recurrence of gastric cancer (log-rank test, *p* = 0.306). The cumulative incidences of mortality were 0% at 3 years and 0% at 5 years in APCA-positive patients; they were 1.52% at 3 years and 2.56% at 5 years in APCA-negative patients. APCA positivity was not significantly associated with 5-year mortality (log-rank test, *p* = 0.371).

### 3.3. Mucin Type, PD-L1 Expression, Immune Cell Infiltration, and Bacteria in Gastric Cancer

There were eight pure AIG patients in the cohort. We matched eight *H. pylori*-positive, APCA-negative patients to eight pure AIG patients by propensity scoring; we investigated mucin type, PD-L1 expression, infiltration of CD3/CD11b/Foxp3/PD1-positive cells, and bacteria in gastric cancer regions by immunohistochemical staining (Table 3 and Table 4 and Figure 3, Figure 4 and Figure 5).

Gastric cancers with a pure AIG background had higher proportions of MUC5AC positivity (8/8 vs. 2/8, *p* = 0.0007) and MUC6 positivity (8/8 vs. 2/8, *p* = 0.0007) than did cancers with *H. pylori* infection.

Gastric cancers with a pure AIG background also tended to have higher expression of PD-L1 than did cancers with *H. pylori* infection (*p* = 0.07). Greater infiltration of CD3-positive (38.13 ± 3.89 vs. 16.50 ± 3.77, *p* = 0.001), Foxp3-positive (14.50 ± 1.56 vs. 5.50 ± 1.02, *p* < 0.001), and PD1-positive (18.38 ± 0.73 vs. 7.25 ± 1.56, *p* = 0.001) cells, as well as lesser infiltration of CD11b-positive (15.38 ± 2.52 vs. 28.13 ± 2.98, *p* < 0.001) cells, were found in the pure AIG group compared with the *H. pylori*-positive group.

The microbiota in the gastric mucosa reportedly differs between AIG and non-AIG patients, likely because of severe gastric acid secretion dysfunction. The abundances of Bacillus and Streptococcus spp. were greater in AIG patients than in non-AIG patients [16]. Based on immunohistological staining, B. cereus abundance was greater in pure AIG gastric cancers than in *H. pylori*-positive cancers (*p* = 0.006), whereas the Streptococcus group B abundances were similar in the two groups (*p* = 1.000).

## 4. Discussion

The findings of this study were as follows: (i) approximately 30% of gastric cancer patients were APCA-positive; (ii) high proportions of MUC6, MUC5AC, and PD-L1 expression, greater infiltration of CD3/Foxp3/PD1-positive cells, and greater B. cereus abundance were associated with pure AIG; (iii) APCA-related gastric cancer tended to have a better prognosis.

Most APCA-positive patients (79.9%) were also *H. pylori*-positive. This finding implies that *H. pylori* infection triggers APCA production [17]. The diagnostic criteria for AIG are not standardized, although most include APCA positivity [18]. New criteria that include anti-intrinsic factor antibody status and other AIG-specific findings are needed.

AIG-related gastric cancer patients had higher levels of MUC6 and MUC5AC expression in the gastric glands. In AIG patients, APCA damages parietal cells but APCA might not damage gastric glands other than parietal cells compared with *H. pylori*-positive gastric cancer. Therefore, molecular carcinogenesis differs between AIG and *H. pylori*-positive gastric cancer. In pure AIG gastric cancer patients, the level of CD3 expression was greater in gastric cancer glands, while the level of CD11 expression was not (Table 2). Tregs are a focus of current research efforts regarding gastric cancer carcinogenesis [19]. Our AIG gastric cancer patients exhibited greater Treg infiltration in gastric cancer glands, compared with non-AIG/*H. pylori*-positive patients (Figure 2). The findings of previous studies support our results [20,21]. This may explain why PD-L1, which suppresses anti-tumor immunity, was highly expressed in AIG-related gastric cancer. PD-1 inhibitors could, thus, have therapeutic potential for AIG-related gastric cancer with a high rate of PD-L1 expression [22].

APCA-positive patients tended to have a better prognosis in terms of 5-year recurrence and mortality, possibly because of MUC6 expression. MUC6 mutation in gastric cancer is associated with a poor prognosis [23]. The downregulation of MUC6 may contribute to malignant transformation of gastric epithelial cells and underlie the growth, invasion, metastasis, and differentiation of gastric carcinoma [23]. Alternatively, the better prognosis of APCA-positive patients might have been caused by the small sample size, or by the low incidence of events. Additionally, the better prognosis of AIG-related gastric cancer might have resulted from different endoscopic follow-up policies and/or detection at an earlier stage. Further studies, particularly using MUC6-knockout in vitro and in vivo models, are warranted.

Streptococcus and Bacillus are specific to the gastric microbiota of AIG-related gastric cancer patients. *B. cereus* abundance was increased in gastric cancer in pure AIG patients. *B. cereus* has been isolated in patients with gastrointestinal cancer; it causes bacteremia and sepsis [24]. Mechanisms such as direct and bacterial metabolite-related tumorigenesis, changes in the stomach microenvironment, and tumor immunity might lead to the induction of specific carcinogenesis. The evidence is limited and further studies of causality using in vitro and in vivo models are needed.

A notable strength of this study was that we evaluated many variables based on medical records; we also performed immunohistochemical staining for MUC5AC, MUC6, PD-L1 CD3, CD11, Foxp3, PD1, and bacteria. Important limitations of this study were its retrospective design and the limited availability of information concerning patient diet, body mass index, and family history. Moreover, the sample size for immunohistochemical staining were very small. Pure AIG gastric cancer were very rare, and it was reported by a previous multi-center study [25]. Further literature analysis is needed in the near future.

## 5. Conclusions

In conclusion, we identified molecular differences between AIG and non-AIG gastric cancer patients. A Treg- and T cell-derived unknown autoimmune system, mucin phenotype, and specific bacteria might be attributed to these pathogeneses.

## Figures and Tables

**Figure 1 biomedicines-10-00884-f001:**
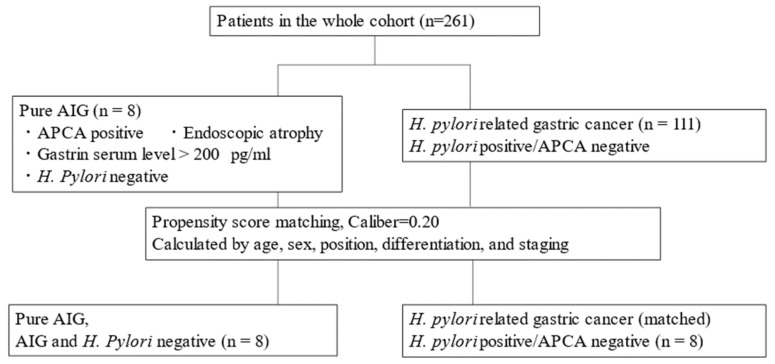
Patient selection flowchart.

**Figure 2 biomedicines-10-00884-f002:**
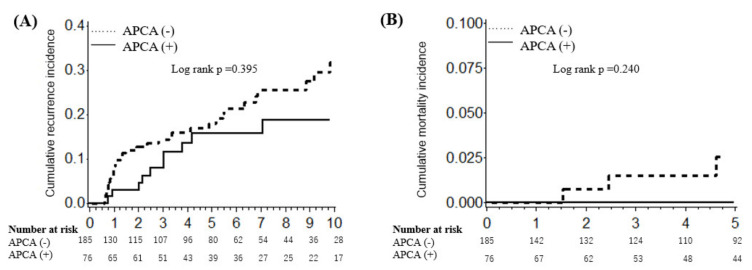
Cumulative incidences of (**A**) gastric cancer recurrence and (**B**) 5-year mortality in APCA-positive vs. APCA-negative patients. Survival analysis was performed using the Kaplan–Meier method and log-rank test. Abbreviation: APCA, anti-parietal cell antibody.

**Figure 3 biomedicines-10-00884-f003:**
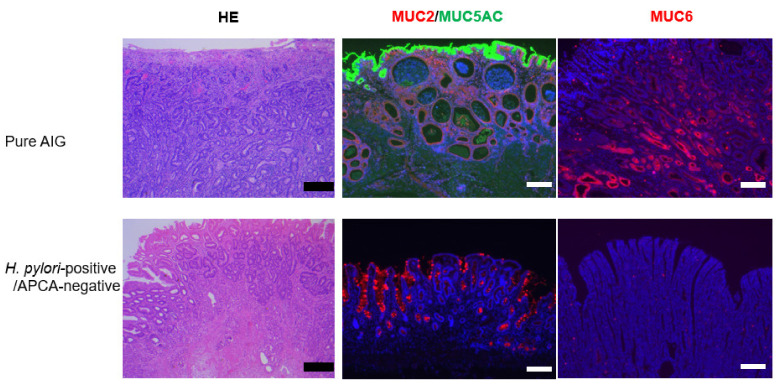
Hematoxylin-eosin and immunohistochemical staining for MUC2, MUC5AC, and MUC6 in gastric cancer in pure AIG and *H. pylori*/APCA-negative patients. High expression of MUC5AC and MUC6 in gastric cancer is associated with pure AIG. Scale bars, 200 μm.

**Figure 4 biomedicines-10-00884-f004:**
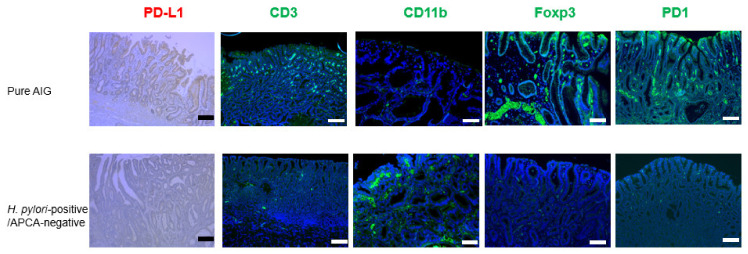
Immunohistochemical staining for PD-L1, CD3, CD11b, Foxp3, and PD1 in gastric cancer in pure AIG and *H. pylori*/APCA-negative patients. Elevated PD-L1 expression and infiltration of CD3/Foxp3/PD1-positive cells in gastric cancer is associated with pure AIG. Scale bars, 200 μm.

**Figure 5 biomedicines-10-00884-f005:**
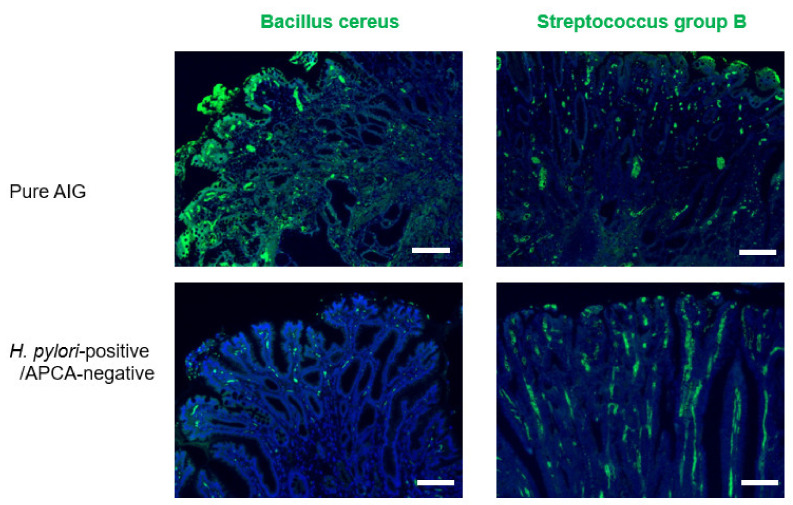
Immunohistochemical staining for bacteria in gastric cancer in pure AIG and *H. pylori*/APCA-negative patients. High abundance of *Bacillus cereus* in gastric cancer is associated with pure AIG. Scale bars, 200 μm.

**Table 1 biomedicines-10-00884-t001:** Baseline patient characteristics.

Characteristic	APCA-Positive (*n* = 76)	APCA-Negative (*n* = 185)	*p* Value
Male/Female	60/16	147/38	0.926
Age (years)	71.39 ± 7.72	67.81 ± 9.60	0.004 *
Gastrin (IU/mL)	421.5 ± 446.9	354.0 ± 406.4	0.238
Atrophic gastritis (non-/closed type/open type)			
Non-	1 (1.32)	4 (2.16)	0.869
Closed type	9 (11.84)	24 (12.97)	
Open type	66 (86.84)	157 (84.86)	
*H. pylori* profile			
*H. pylori*-positive (from patient history, histology, or antibody)	12 (15.79)	29 (15.68)	0.505
*H. pylori* eradicated	38 (50.00)	82 (44.32)	
*H. pylori-*negative	15 (19.74)	32 (17.30)	
*H. pylori* status unknown	11 (14.47)	42 (22.70)	
Comorbidities			
Ischemic heart diseases	10 (13.16)	27 (14.59)	0.762
Chronic heart failure	18 (23.68)	25 (13.51)	0.044 *
Peripheral vascular diseases	3 (3.95)	7 (3.78)	0.950
Cerebrovascular diseases	7 (9.21)	11 (5.95)	0.344
Dementia	4 (5.26)	4 (2.16)	0.187
Chronic obstructive pulmonary disease	5 (6.58)	7 (3.78)	0.327
Rheumatoid diseases	5 (6.58)	4 (2.16)	0.076
Peptic ulcer diseases	48 (63.16)	110 (59.46)	0.579
Diabetes mellitus	29 (38.16)	65 (35.14)	0.644
Chronic kidney diseases	1 (1.32)	9 (4.86)	0.175
Hemiplegia	3 (3.95)	3 (1.62)	0.255
Leukemia	0 (0.00)	0 (0.00)	1
Lymphoma	3 (3.95)	3 (1.62)	0.255
Liver cirrhosis	7 (9.21)	18 (9.73)	0.897
Acquired immunodeficiency syndrome	0 (0.00)	0 (0.00)	1
Medications			
Aspirin	8 (10.53)	19 (10.27)	0.951
Proton pump inhibitors	38 (50.00)	78 (42.16)	0.247

Abbreviations: AIG, autoimmune gastritis; APCA, anti-parietal cell antibody. Number of patients (%) or mean ± standard deviation. *: *p* values < 0.05 were considered to be significant.

**Table 2 biomedicines-10-00884-t002:** Gastric cancer characteristics.

Characteristic	APCA-Positive (*n* = 76)	APCA-Negative(*n* = 185)	*p* Value
Tumor location			
Antrum	52 (68.42)	133 (71.89)	0.575
Corpus/cardia	24 (31.58)	52 (28.11)	
Differentiation			
tub1/tub2	59 (77.63)	147 (79.46)	0.742
por/sig	17 (22.37)	38 (20.54)	
Epstein–Barr virus-positive	2 (2.63)	3 (1.62)	0.630
Staging			
pstage (III or IV = 1)	4 (5.26)	28 (15.14)	0.027 *
pT (III or IV = 1)	5 (6.58)	29 (15.68)	0.067
pN (I~ = 1)	5 (6.58)	29 (15.68)	0.067
pM (I = 1)	3 (3.95)	15 (8.11)	0.290
Initial treatment			
Surgical resection	24 (31.58)	63 (34.05)	0.700
Endoscopic resection	50 (65.79)	112 (60.54)	0.427
Chemotherapy	1 (1.32)	6 (3.24)	0.381
Best supportive care	1 (1.32)	4 (2.16)	0.650

Abbreviation: AIG, autoimmune gastritis. *: *p* values < 0.05 were considered to be significant.

**Table 3 biomedicines-10-00884-t003:** Baseline characteristics of the matched cohort.

Characteristic	Pure AIG (*n* = 8)	*H. pylori*-Positive/APCA-Negative (*n* = 8)	*p* Value
Male/Female	6/2	7/1	1.00
Age (years)	75.38 ± 4.10	76.13 ± 4.52	0.733
APCA-positive	8 (100.00)	0 (0.00)	0.0002 *
Gastrin (IU/mL)	550.0 ± 563.2	462.5 ± 309.5	0.706
Atrophic gastritis (non-/closed type/open type)			
Non-	0 (0.00)	0 (0.00)	1.00
Closed type	1 (12.50)	0 (0.00)	
Open type	7 (87.50)	8 (100.00)	
*H. pylori* profile			
*H. pylori*-positive (from patient history, histology, or antibody)	0 (0.00)	2 (25.00)	0.0002 *
*H. pylori* eradicated	0 (0.00)	6 (75.00)	
*H. pylori-negative*	8 (100.00)	0 (0.00)	
Tumor location			
Antrum	2 (25.00)	5 (62.50)	0.315
Corpus/cardia	6 (75.00)	3 (37.50)	
Differentiation			
tub1/tub2	7 (87.50)	5 (62.50)	0.248
por/sig	1 (12.50)	3 (37.50)	
Epstein–Barr virus-positive	0 (0.00)	0 (0.00)	1
Staging			
pstage (III or IV = 1)	0 (0.00)	0 (0.00)	1
pT (III or IV = 1)	0 (0.00)	0 (0.00)	1
pN (I~ = 1)	0 (0.00)	0 (0.00)	1
pM (I = 1)	0 (0.00)	0 (0.00)	1
Initial treatment			
Surgical resection	3 (37.50)	3 (37.50)	1
Endoscopic resection	5 (62.50)	5(62.50)	1
Chemotherapy	0 (0.00)	0 (0.00)	1
Best supportive care	0 (0.00)	0 (0.00)	1

Number of patients (%) or mean ± standard deviation. *: *p* values < 0.05 were considered to be significant.

**Table 4 biomedicines-10-00884-t004:** Mucin type, immune cell numbers, PD-L1 expression, and bacteria.

Characteristic	Pure AIG (*n* = 8)	*H. pylori-*Positive/APCA-Negative (*n* = 8)	*p* Value
**Mucin type**			
MUC2-positive	4 (50%)	8 (100%)	0.07
MUC5AC-positive	8 (100%)	2 (25%)	0.007 *
MUC6-positive	8 (100%)	2 (25%)	0.007 *
**PD-L1 expression**			
TPS > 50%	4 (50%)	2 (25%)	0.07
TPS 1–49%	4 (50%)	2 (25%)	
TPS < 1%	0 (0%)	4 (0%)	
**Immune cell number**			
CD3-positive cells	38.13 ± 3.89	16.50 ± 3.77	0.001 *
CD11b-positive cells	15.38 ± 2.52	28.13 ± 2.98	0.006 *
Foxp3-positive cells	14.50 ± 1.56	5.50 ± 1.02	<0.001 *
PD1-positive cells	18.38 ± 0.73	7.25 ± 1.56	<0.001 *
**Bacteria**			
*Bacillus cereus*			
Grade 1 (mild)	0 (0%)	5 (62.5%)	0.007 *
Grade 2 (moderate)	4 (50%)	3 (37.5%)	
Grade 3 (severe)	4 (50%)	0 (0%)	
*Streptococcus* group B			
Grade 1 (mild)	3 (37.5%)	3 (37.5%)	1
Grade 2 (moderate)	5 (62.5%)	4 (50%)	
Grade 3 (severe)	0 (0%)	1 (12.5%)	

Abbreviation: AIG, autoimmune gastritis. *: *p* values < 0.05 were considered to be significant.

## Data Availability

All data presented in this study are included within the paper and its Appendix A.

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
