# Peer review of "Clinicopathological Features of Gastric Cancer with Autoimmune Gastritis"

_biomedicines, 2022, doi:10.3390/biomedicines10040884_

Round 1

Reviewer 1 Report

The objective of the authors was to analyze the differences in clinicopathological features of gastric cancer in patients with helicobacter pylori induced gastritis and autoimmune gastritis (AIG). It is retrospective cohort study on patients hospitalized in the University Tokyo hospital from 1998-2017 with gastric cancer diagnosis. The authors have evaluated mucin expression, immune cell infiltration, gastric microbiota. The authors have also performed gastric cancer recurrence and incidence rates of mortality in AIG and non-AIG gastric cancer patients. The authors have reported increased expression of MUC5AC, MUC6, PD-L1, and higher infiltration of CD3/FoxP3/PD-1 positive cells in pure AIG related gastric cancer patients. The authors have also reported APCA positivity was not significantly associated with gastric cancer recurrence and cumulative incidence rates of mortality. It is interesting data but the manuscript could not be considered for publication in the current form for following reasons.

Major concerns:

1) The major drawback is the sample size and gender. The pure AIG group has a sample size of n=8. The pure AIG group has a sample size of n=2 and n=1 for matched non-AIG group for gender (female) type. Is there a way for authors to increase sample size even if it meant to include samples from other hospitals to address this issue?

2) Could authors carefully reanalyze the data reported in Table 1. There are several concerns as listed below.
2A) The authors have denoted the percentage for H. Pylori eradicated group as n=38 (50) and n=82 (44.32) for APCA positive and negative groups. Could authors please clarify what factors (total sample size or H. pylori was eradicated in patients reported as positive) the authors have considered to come up with the above mentioned number/conclusion? The authors are requested to provide the clarity and details accordingly.
2B) The total sample size do not match when we combine sub groups sample size for data mentioned under H. pylori profile?
2C) If the baseline characteristics mentioned in Table-1 is cumulative representative of total group (Male + Female); the authors must include sample size for gender type as well for clarity.

3) The authors have stated that two different time period for this retrospective study. In the abstract the authors have stated this study was based on data collected from gastric cancer patients admitted at Tokyo University Hospital from 1998-2017 but in material and methods section the authors have stated the timeline as 1996-2017. Could authors please crosscheck and fix the issue.

Author Response

The objective of the authors was to analyze the differences in clinicopathological features of gastric cancer in patients with helicobacter pylori induced gastritis and autoimmune gastritis (AIG). It is retrospective cohort study on patients hospitalized in the University Tokyo hospital from 1998-2017 with gastric cancer diagnosis. The authors have evaluated mucin expression, immune cell infiltration, gastric microbiota. The authors have also performed gastric cancer recurrence and incidence rates of mortality in AIG and non-AIG gastric cancer patients. The authors have reported increased expression of MUC5AC, MUC6, PD-L1, and higher infiltration of CD3/FoxP3/PD-1 positive cells in pure AIG related gastric cancer patients. The authors have also reported APCA positivity was not significantly associated with gastric cancer recurrence and cumulative incidence rates of mortality. It is interesting data but the manuscript could not be considered for publication in the current form for following reasons.

Major concerns:

1) The major drawback is the sample size and gender. The pure AIG group has a sample size of n=8. The pure AIG group has a sample size of n=2 and n=1 for matched non-AIG group for gender (female) type. Is there a way for authors to increase sample size even if it meant to include samples from other hospitals to address this issue?

Response: Unfortunately, we could not increase the pure AIG gastric cancer group (n=8). However, the pure AIG gastric cancer patients were very rare cases. In recent multi-center study, it included only 6 AIG gastric cancer patients with APCA positive [United European Gastroenterol J 2020; 8: 175–184]. We have clarified that in the limitation.

2) Could authors carefully reanalyze the data reported in Table 1. There are several concerns as listed below.

2A) The authors have denoted the percentage for H. Pylori eradicated group as n=38 (50) and n=82 (44.32) for APCA positive and negative groups. Could authors please clarify what factors (total sample size or H. pylori was eradicated in patients reported as positive) the authors have considered to come up with the above mentioned number/conclusion? The authors are requested to provide the clarity and details accordingly. The total sample size do not match when we combine sub groups sample size for data mentioned under H. pylori profile?

Response: Thank you for your comments. We have corrected the number of patients of H. pylori profile.

2B) If the baseline characteristics mentioned in Table-1 is cumulative representative of total group (Male + Female); the authors must include sample size for gender type as well for clarity.

Response: We have added the number of female patients in Table 1.

3) The authors have stated that two different time period for this retrospective study. In the abstract the authors have stated this study was based on data collected from gastric cancer patients admitted at Tokyo University Hospital from 1998-2017 but in material and methods section the authors have stated the timeline as 1996-2017. Could authors please crosscheck and fix the issue.

Response: Thank you. We have corrected as “1998-2017” in the revised manuscript.

Reviewer 2 Report

the work is of good quality, I suggest the following:

1. Table 1 is very large, can you divide it into two parts? Thank you
2. in table 1, only age had a significant difference? What is the importance of this table?
3. If you compare the values between each group, what data do they give you? APCA-positive group, and APCA-negative group. For example, if among the APCA-positive, the statistics are made between men and women, are there differences?
4. As the data is presented in Tables 1 and 2, no statistical differences are observed. Should you do another type of analysis?
5. Can you explain why there is a male predominance? what are the female sex data? compared female vs male? in both groups (APCA-positive group, and APCA-negative group)
6. Did they carry out optical microscopy studies?

Author Response

the work is of good quality, I suggest the following:

  1. Table 1 is very large, can you divide it into two parts? Thank you

Response: Thank you for your comments. We have moved the Table regarding medications except for PPI and aspirin to the Supplementary Table1.

  1. in table 1, only age had a significant difference? What is the importance of this table?

Response: Previously, the clinical patient characteristics of APCA positive and negative gastric cancer did not fully elucidated. In this study, we found that only age had a significant difference between the groups. We think that these results were new valuable findings.

  1. If you compare the values between each group, what data do they give you? APCA-positive group, and APCA-negative group. For example, if among the APCA-positive, the statistics are made between men and women, are there differences?

Response: Main purpose of comparison of patient characteristics between APCA-positive and APCA negative was to evaluate the balance of recurrence risk. If APCA-positive cancer patient group had higher recurrence risk such as advanced atrophic gastritis, higher proportion of immunosuppressed diseases, and lower proportion of chemo preventive aspirin use, we need to adjust these confounder risks using statistical model such as cox proportional hazard model. As a result of baseline recurrence risk of gastric cancer between the group in Table 1 and 2, patient characteristics was principally well balanced between the groups. Thus, we compared the cumulative recurrence incidence using the Kaplan–Meier method.

  1. As the data is presented in Tables 1 and 2, no statistical differences are observed. Should you do another type of analysis?

Response: We think that there is no need to perform further analysis to evaluate baseline recurrence cancer risk. Almost baseline patient characteristics including gastrin level, atrophic gastritis, H. pylori profile, various comorbidities, and potential chemo preventive drugs was well balanced between the groups.

  1. Can you explain why there is a male predominance? what are the female sex data? compared female vs male? in both groups (APCA-positive group, and APCA-negative group)

Response: Thank you for thoughtful suggestion. We did not fully understand why male patient was predominant in APCA positive and negative group. One possible explanation is that we extracted gastric cancer patient data from our cohort. Generally, gastric cancer was male predominant. In our data, male consists approximately 80%. Thus, male patient was predominant in both groups. We have added the number of female patients in Table 1.

Next, we performed additional analyses to evaluate an association between sex data and gastric cancer recurrence. No significant differences in the recurrence was observed between male and female sex in APCA positive and APCA negative patients.

  1. Did they carry out optical microscopy studies?

Response: Yes, we have performed optical microscopy studies. We have added these figures (Figure 3).

Round 2

Reviewer 1 Report

The authors have addressed the comments and the manuscript could be considered for publication.

Reviewer 2 Report

no